# Infoveillance and Critical Analysis of the Systematically Reviewed Literature on Dimethyltryptamine and the “God Molecule”

**DOI:** 10.3390/ph16060831

**Published:** 2023-06-02

**Authors:** Ahmed Al-Imam, Marek A. Motyka, Beata Hoffmann, Anita Magowska, Michal Michalak

**Affiliations:** 1Department of Computer Science and Statistics, Doctoral School, Poznan University of Medical Sciences, 61-806 Poznan, Poland; 2Department of Anatomy and Cellular Biology, College of Medicine, University of Baghdad, Baghdad 10047, Iraq; 3Barts and the London School of Medicine and Dentistry, Queen Mary University of London, London E1 2AD, UK; 4Institute of Sociological Sciences, University of Rzeszow, 35-959 Rzeszów, Poland; mmotyka@ur.edu.pl; 5Institute of Applied Social Sciences, University of Warsaw, 00-927 Warsaw, Poland; beata.hoffmann@uw.edu.pl; 6Department of History and Philosophy of Medical Sciences, Poznan University of Medical Sciences, 61-806 Poznan, Poland; vesalius@ump.edu.pl; 7Department of Computer Science and Statistics, Poznan University of Medical Sciences, 61-806 Poznan, Poland; michal@ump.edu.pl

**Keywords:** 5-MeO-DMT, 5-methoxy-*N*,*N*-dimethyltryptamine, Colorado river toad, *Incilius alvarius*, psychedelic agents, Sonoran Desert toad

## Abstract

Aboriginals of Latin America have used DMT (*N*,*N*-dimethyltryptamine) in ritualistic ceremonies for centuries. Nevertheless, there are limited data on web users’ interest concerning DMT. We aim to review the literature and explore the spatial–temporal mapping of online search behavior concerning DMT, 5-MeO-DMT, and the Colorado River toad via Google Trends over the past 10 years (2012–2022) while using 5 search terms: “*N*,*N*-dimethyltryptamine”, “5-methoxy-*N*,*N*-dimethyltryptamine”, “5-MeO-DMT”, “Colorado River toad”, and “Sonoran Desert toad”. Literature analysis conveyed novel information concerning DMT’s past shamanic and present-day illicit uses, showcased experimental trials on DMT uses for neurotic disorders, and highlighted potential uses in modern medicine. DMT’s geographic mapping signals originated mainly from Eastern Europe, the Middle East, and Far East Asia. In contrast, 5-MeO-DMT signals prevailed in Western Europe, Indo-China, and Australasia. Signals concerning the toad originated from the Americas, Australia, India, the Philippines, and Europe. Web users searched the most for “*N*,*N*-dimethyltryptamine” and “5-MeO-DMT”. Three terms exhibited significant upgoing linear temporal trends: “5-MeO-DMT” (β = 0.37, *p* < 0.001), “Sonoran Desert toad” (β = 0.23, *p* < 0.001), and “Colorado River toad” (β = 0.17, *p* < 0.001). The literature and Infoedemiology data provided crucial information concerning DMT’s legal status, risks and benefits, and potential for abuse. Nonetheless, we opine that in the upcoming decades, physicians might use DMT to manage neurotic disorders pending a change in its legal status.

## 1. Introduction

Dimethyltryptamine (DMT) is an indole alkaloid and a derivative of tryptamine (a metabolite of the essential amino acid tryptophan), which exists in many biological systems (endogenous substance) including plants and animals [1,2]. DMT produces intense but short-lived (up to 30–60 min) psychedelic and hallucinogenic experiences [1]. There is also increasing evidence that endogenous DMT plays a role in the central (CNS) and peripheral nervous system (PNS), and it may act as a neurotransmitter by acting as a non-selective serotonin agonist principally on the 5-hydroxytryptamine (5-HT1A and 5-HT2A) receptors [1,2]. Serotoninergic psychedelics, including DMT, can boost mood, social interaction, and cognition when used in small doses, intermittently, and in the long term; the former administration method is known as micro-dosing [3]. There is also evidence that the human body synthesizes DMT, most likely within the epiphysis cerebri (pineal gland) [4].

The results of studies conducted in the last decade on the extent of DMT and 5-MeO-DMT use are imprecise. Some data suggest that these substances are used infrequently [5]. Others suggest that the use of these psychedelics is increasing slightly [6]. Global use estimates of 5-MeO-DMT are challenging to determine due to the omission of data on the use of these psychedelics in global epidemiological diagnoses. 5-MeO-DMT is most often classified under “new psychoactive substances”, which limits the ability to estimate its global prevalence of use [7]. Admittedly, the Global Drug Survey 2021 reported and presented percentage data on DMT use, but this research was not conducted on a representative sample, so the reported results should not be considered a general trend [8]. Moreover, the prevalence of DMT use is ambiguous worldwide and can vary substantially from one country to another, pending its availability.

The 5-methoxy-*N*,*N*-dimethyltryptamine (5-MeO-DMT) is a pharmacodynamically singular variant (derivative) of DMT, and it is one of the most potent psychedelics [9,10]. Both DMT and 5-MeO-DMT can be administered using micro-dosing. Researchers further define micro-dosing as regular but intermittent ingestion of minute doses of psychedelics to improve well-being, performance, cognition, mood, or interpersonal processes [3,11]. The 5-MeO-DMT exists in several plant species and the Colorado River toad (*Incilius alvarius*; Girard); the toad is native to northern Mexico and the southwestern United States [9,10]. The indigenous inhabitants of Latin America also used it for spiritual purposes, during which they experienced radical alteration of consciousness and mystical experiences [10,12]. Ermakova et al. (2022) reported that 5-MeO-DMT is a serotonergic agonist with the highest affinity for 5-HT1A receptors. However, these data relate to pre-clinical (animal) studies [12], while human studies are lagging due to ethical and medico-legal constraints because DMT and other psychoactive substances are categorized by the United Nations as controlled chemicals under the category of “Schedule I drugs” [13].

Historically, scholars believe that Native Americans used the 5-MeO-DMT—also known as the “God Molecule”—for spiritual purposes, especially in shamanic practices [3,11]. Hoshino and Shimodaira were the first to synthesize it in a laboratory in 1936 [14]. According to Tyler and Gröger (1964), 5-MeO-DMT is also found in fungi [15], while Erspamer and coworkers (1967) detected it in the venom of the glandular secretions of the Colorado River toad, formerly known as the Sonoran Desert toad [16]. Furthermore, 5-MeO-DMT is the primary psychoactive substance in ayahuasca, a psychedelic tea originating from the aboriginal inhabitants of the Amazonian basin, since at least 3000 years ago, and it was frequently consumed by ancestral inhabitants of current-day Bolivia, Ecuador, Peru, Venezuela, and Brazil during ceremonies for centuries [17,18,19]. In modern times, the toad venom extract and the synthetic powder form are used for psychedelic experiences, attracting much tourism to these Latin nations of Central and South America (Figure 1) [20].

An anecdotal hypothesis related to the history of DMT is the “*Stoned Ape Theory*”, which is not backed up by evidence. It claims that psychedelics, including psilocybin and DMT, were crucial catalysts for early hominids’ evolution via the augmentation of the cranial capacity, thinking capabilities, and the emergence of higher cognitive faculties, including language, manual dexterity, and problem-solving [21]. On the other hand, evolutionary biologists confirmed specific genes’ role in enhancing human cognition throughout the evolutionary timeline [22,23]. For example, the FOXP2 gene promotes neuronal synaptic plasticity, increases dendrites’ connectivity, and enhances the brain’s cortico-basal ganglia circuits [22]. Furthermore, the appearance and stabilization of the FOXP2 gene coincided with the first presence and quick spread of Homo sapiens [23]. The FOXP2 and similar genes could better explain modern humans’ (*Homo sapiens*) superior cognition, yet the interaction between the former genomic machinery and serotoninergic psychedelics is still unknown. In other words, future research can provide evidence of whether psychedelics could possess mutagenic properties influencing the genome.

As Hong-Wu Shen et al. pointed out, 5-MeO-DMT is a potent, fast-acting hallucinogen with a short duration in humans. Psychedelic effects occur after various routes of administration, including inhalation (~6–20 mg), intravenous injection (~0.7–3.1 mg), and oral administration (~30 mg; with MAO inhibitor). The effects of use begin after 3–4 min, climax around 35–40 min, and subside after about 60–70 min [19]. Experiments conducted with humans have shown that 5-MeO-DMT causes profound psychedelic effects, including drastic changes in consciousness and many subjective manifestations. Subjective experiences include: (a) physical effects (tactile enhancement, tactile hallucinations, muscular irritability, tremors, skin flushing, pupillary dilation, nausea, respiratory depression, spatial disorientation, and physical autonomy); (b) visual phenomena (visual acuity enhancement or suppression, color shifting, and visual hallucinations); (c) cognitive effects (amnesia, anxiety, ego dissolution (ego death), confusion, delusions, emotions’ intensification, time perception distortions, and increased wakefulness); (d) auditory phenomena (autonomous voice communication, auditory enhancement, auditory distortions, and hallucinations); (e) and trans-personal effects (perception of eternalism, pre-determinism, and interconnectedness with the universe) [24].

### Aim and Rationale of the Study

The current study aims to review the literature concerning DMT’s past (historical) uses, legal aspects, clinical manifestations, subjective experiences, incidents of intoxications and fatalities, uses in modern medicine, and potential applications in personalized medicine. The current research will also explore the infodemiology of DMT over the Internet. Infodemiology could complement classical epidemiology in describing phenomena of interest, including substance misuse and addictive behavior. There might be a correlation between actual substance use and Internet search behavior concerning the abused substance. Therefore, our study also aims to examine the online information search (seeking) behavior, also known as the search interest, of web users of the Internet (surface web).

We will extrapolate inferential data concerning worldwide search interest’s spatial (geographic) and temporal (time-series) mapping. The mapping can be crucial to contrast real-life epidemiology with digital-based infodemiology data concerning the use and interest (curiosity) in DMT, 5-MeO-DMT, and other relevant topics, such as the Colorado River toad. On the other hand, temporal mapping can provide information on when a particular substance is sought throughout the year and which substance is the most popular, while geographic mapping could indicate geographic foci (regions) of higher interest concerning a specific substance, such as DMT. The former information concerning search behavior might help to predict actual substance use behavior, which could be crucial for policymakers and health-care providers. To the best of the authors’ knowledge and per the systematic review of the published literature, the current study is the first research concerning the infodemiology of DMT while exploring additional domains, including historical data and uses in modern and personalized medicine.

## 2. Results

### 2.1. The Uses of DMT in Past and Modern Times

According to Eckernäs et al. (2022), DMT is a psychedelic that can potentially cure several psychiatric disorders, although little is known about its pharmacokinetics and pharmacodynamic properties [25]. Nevertheless, the lack of knowledge concerning DMT and its properties did not prevent people from the ancient past and modern times from experimenting with it at their discretion. Research data confirm that one of the more popular decoctions containing DMT is ayahuasca, consumed during so-called rites of passage in the Amazon since pre-Columbian times, whereby its use and sensations were monitored during these ceremonies by tribal chiefs who thought that these experiences were crucial in preparing young people for the roles assumed by adults [26].

The base for making this decoction is *Banisteriopsis caapi* (Spruce ex Griseb.) C.V.Morton, a plant that grows over vast South American terrains, and which can be found in the eastern part of Brazil, Ecuador, Bolivia, Peru, Colombia, and the Caribbean. It owes its Latin name to the missionary and botanist John Banister, who roamed the Amazon Forest in the 17th century. The psychoactive constituents to which the liana owes its appeal are the β-carboline alkaloids tetrahydroharmine, harmaline, and harmine. Taken in small doses, they exert an antidepressant effect. However, in larger doses, they induce a hallucinogenic experience. A hallucinogenic decoction is also produced when combined with DMT-containing plants, such as *Psychotria viridis* Ruiz & Pav. [27]. Ayahuasca accompanied indigenous people at nuptials, birthdays, and initiations in the Amazon region. At the same time, ayahuasca is used in natural (traditional or cultural) medicine for strengthening, aiding concentration and performance, and cleansing the body of accumulated toxins. Among indigenous Amazonian peoples, any circumstances during which it is permissible to use the plant for medicinal and ceremonial purposes are regulated by internal tribal arrangements [28]. Both the preparation and serving of the brew are only allowed to curanderos (local healers—shamans) [27].

Some researchers suggest that the DMT-containing secretion of the toad *Incilius alvarius* may also have been used in the southwestern U.S. and in northern Mexico. It has been established that a single toad can produce doses sufficient to achieve psychedelic sensations. Admittedly, the doses are toxic when ingested orally, but burning the collected secretions followed by inhalation produces potent psychedelic results [29].

In Western culture, it is possible to identify several important dates relating to the modern history of DMT. According to Steven Barker, such a date is 1931, when Canadian chemist Richard Manske conducted the first synthesis of DMT in a laboratory [4]. Barker also pointed out that the occurrence of DMT in plants was discovered in 1946 by microbiologist Oswaldo Gonçalves de Lima, while the psychedelic properties of DMT were described in 1956 by Hungarian chemist and psychiatrist Stephen Szár, who extracted DMT from the Mimosa plant and conducted an experiment on himself by taking the extract via intramuscular injection [4]. One can also find slightly different dates for these discoveries in the scientific literature. According to Anna Ermakova et al., the first synthesis of 5-MeO-DMT took place in 1936, while the first isolation of the psychedelic from the *Dictyoloma incanescens* DC. plant of the *Rutaceae* family took place in 1959, and in subsequent years, it was identified in some fungi, secretions of the glands of the desert toads, and in mammals [12]. Despite the slight discrepancies in time, it must be acknowledged that these events bridge the gap between the profane and the sacred and between modern science and the cultural–religious use of many DMT-containing plants [4].

The use of DMT for ritual purposes is also observed in modern times. Indigenous South Americans have been using plants containing 5-MeO-DMT for thousands of years [30], and they used snuff made from *Anadenanthera peregrina* (L.) Speg. beans [12]. In Brazil, until modern times, there are several religious communities for which ayahuasca decoction is a sacrament in public rituals. Congregations celebrating ayahuasca are also found in Germany, Australia, Canada, France, the Netherlands, Japan, Spain, and the United States [31,32]. In the 21st century, ayahuasca healing sessions conducted by globetrotting curanderos are increasingly common and available to almost anyone interested [27]. As for the use of DMT-containing secretions of *Incilius alvarius* toads, researchers could not find definite historical evidence of its indigenous use, and it could be a relatively recent phenomenon [12].

Today, researchers are exploring the medicinal properties of psychedelics and providing a better understanding of their mood-enhancing and potential therapeutic properties [33]. At least a few studies have been undertaken in the last decade. Ido Hartogsohn, examining a cult originating in Brazil that uses ayahuasca brew during rituals, describes the profound dependence of psychedelic effects on psychological, social, and cultural factors [33,34]. Deborah Gonzalez et al. (2021) tracked data collected from 200 patients with psychiatric disorders treated with ayahuasca decoction, all of whom experienced marked and sustained improvements in health, quality of life, and spirituality [35].

In the most extensive study reported to date, collected from nearly 7000 respondents consuming ayahuasca, valuable data demonstrated the vital role of combining the ceremonial context of decoction use, therapeutic motivations, and the additional support obtained with yoga and tai chi exercises (Tai chi ch’üan), also sometimes known as “shadowboxing”, to produce positive results in terms of improved well-being, mystical experiences, and insight [36]. These and other studies suggest that psychedelic experiences correspond closely with the enhancement and transcendence of mood, spirituality, cognitive abilities, and quality of life.

### 2.2. Legal Aspects

We should note two normative approaches when considering the legal aspects. The first is regulated by state or international laws, which aim to protect against the widespread use of DMT-containing specifics and reduce possible adverse events resulting from uncontrolled use. The second aspect is ritual–cultural use, which is regulated by cultural precepts but in a less restrictive manner.

DMT is on Schedule One of the 1971 United Nations Convention on Psychotropic Substances, which means that the drug is subject to mild restrictions, its use has no therapeutic effect, and there is a high potential for dependence, abuse, and the possibility of severe adverse effects [13]. However, in 21st-century Brazilian syncretic religions, ayahuasca use has spread to almost every continent; in several countries, these groups have obtained some form of legal authorization for the ritual and religious use of ayahuasca [37]. Congregations celebrating ayahuasca exist in Germany, Australia, Canada, France, the Netherlands, Japan, Spain, and the United States [31,32,38]. In contrast, the United Nations indicates that DMT can be used strictly for medical research and scientific purposes, while international trade is under vigilant and systematic observation [39].

Synthetic DMT derivatives appeared with the escalation of the so-called “legal highs” on the drug market—especially on the deep web and the darknet—influencing the introduction of appropriate “legal” arrangements [40]. In Poland, in March 2009, as part of an amendment to the Act on Counteracting Drug Addiction of 29 July 2005, the plants from which ayahuasca decoction is prepared were placed on the list of narcotics, and their distribution and use were subject to legal sanctions [41]. The 5-MeO-DMT is a controlled substance in New Zealand, the U.K., Australia, and several other countries [12].

As for the ritual–cultural aspects, the situation is not subject to overly restrictive sanctions. For this reason, ceremonies during which DMT-containing hallucinogenic decoctions are consumed could be said to promote the undertaking of travel to these places to experience mystical and entheogenic states [27]. In recent decades, so-called ayahuasca tourism has become popular [32]. Reported reasons for tourists reaching for ayahuasca include curiosity, treatment of mental disorders, the need for self-discovery, spiritual development, interest in psychedelic medicine, the search for purpose in life, and the opportunity to contact God, spirits, and transcendental energy, as well as hedonistic goals [42,43]. Of course, it should be noted that the exact specifics that can serve to improve psycho–physical–social well-being, when used solely for recreational purposes, can at the same time pose a critical threat to users [44]. Thus, it should be noted that true shamans understand and respect the powers hidden in plants, and none use hallucinogenic plants for entertainment, as Westerners do [45].

### 2.3. Google Trends: Infodemiology and Infoveillance

#### 2.3.1. Geographic (Spatial) Mapping

Concerning the geographic mapping of the search term “*N*,*N*-dimethyltryptamine”, the top twenty countries included Sri Lanka, Slovenia, Lithuania, Israel, Bulgaria, Serbia, Egypt, South Korea, Iran, Turkey, Morocco, Pakistan, Saudi Arabia, United Arab Emirates, Australia, Austria, Poland, Greece, India, and Indonesia. Most of these countries are from Eastern Europe, the Middle East, and Far East Asia. The search term “5-methoxy-*N*,*N*-dimethyltryptamine” never generated spatial mapping data. On the contrary, the abbreviated “5-MeO-DMT” yielded a thorough and distinct mapping, and the top contributing countries included Denmark, Vietnam, China, Japan, Norway, Canada, the Netherlands, Portugal, Australia, South Africa, Poland, Sweden, Thailand, Germany, Romania, Malaysia, Austria, New Zealand, Ireland, and India. The former nations were mainly from Western Europe, Indo-China, and Australasia (Figure 2).

Concerning the two terms related to *Inciliius alvarius* species, the geographic mapping (Figure 2) was relatively limited. The top nations that mapped the “Colorado River toad” included Uruguay, Costa Rica, Paraguay, Panama, Guatemala, Peru, Ecuador, Venezuela, Spain, Colombia, Mexico, Argentina, Brazil, Chile, Italy, Czechia, Russia, Slovakia, and Greece. These are mainly Latin countries from South and Central America, Eastern European countries, and a few northern Mediterranean countries. Signals concerning the search term “Sonoran Desert toad” were limited to nine countries only, including the Philippines, United States, India, United Kingdom, Australia, Canada, Italy, Germany, and the Netherlands; these represent nations from North America, Australasia, the Indian subcontinent, and Western Europe. Data signals from Arab countries and the Middle East only concerned the search term “*N*,*N*-dimethyltryptamine”, while those from Poland were mapped to almost each search term. The Middle Eastern and Arab countries included Saudi Arabia, Morocco, Egypt, Turkey, Iran, Palestine, and the United Arab Emirates. The former nations accounted for almost one-eighth (12.07%) of the geographic map [46].

#### 2.3.2. Temporal Mapping

Concerning the temporal mapping, the online information search behavior indicated that web users searched the most for “*N*,*N*-dimethyltryptamine” (range: 55 to 100), and to a lesser extent for “5-MeO-DMT” (1 to 17), “Colorado River toad” (1 to 3), and “Sonoran Desert toad” (0 to 1). In contrast, they searched the least for “5-methoxy-*N*,*N*-dimethyltryptamine” (0 to 0.5) (Figure 3). Besides, the search term “*N*,*N*-dimethyltryptamine” exhibited statistical outliers during the first quarter of 2019 (January and March) and the second quarter of 2020 (May), while “5-MeO-DMT” had only one outlier at the beginning of 2021 (January). On the other hand, there were no outliers concerning the “Colorado River toad” search term. However, the “Sonoran Desert toad” had some outliers that occurred during the third quarter of 2019 (August), the fourth quarter of 2021 (September and November), and the second half of 2022 (July, August, and November). On the other hand, the least popular search term (“5-methoxy-*N*,*N*-dimethyltryptamine”) had frequent outliers throughout the past nine years (2013–2022). The outliers that existed around June 2020 belonged to two search terms (“*N*,*N*-dimethyltryptamine” and “5-methoxy-*N*,*N*-dimethyltryptamine”). The former outlier (during June 2020) might partially relate to the news in association with the article the BBC published in early June 2020, titled: “*Porn star Nacho Vidal held in Spain after man dies in toad-venom ritual*”. The Spanish police reported that a photographer died after inhaling the poison of a North American toad during a “mystic ritual” in Vidal’s home [47].

Concerning the nature of the linear trend for each search term, all the terms had positive linear trends (Table 1). Nevertheless, only three search terms had a significant linear relationship with time, including “5-MeO-DMT” (β = 0.37, *p* < 0.001), “Sonoran Desert toad” (β = 0.23, *p* < 0.001), and “Colorado River toad” (β = 0.17, *p* < 0.001). The former search terms are the same that exhibited significant bivariate correlations. Furthermore, the correlation matrix revealed significant correlations of moderate-to-large effect sizes. These correlations indicate that web users’ collective behavior searched the most for three related terms: “5-MeO-DMT”, “Colorado River toad”, and “Sonoran Desert toad”.

## 3. Discussion

### 3.1. Highlights of the Most Important Findings

The geographic mapping signals concerning DMT originated mainly from Eastern Europe, the Middle East, and Far East Asia, while those concerning its most potent variant (5-MeO-DMT) came from Western Europe, Indo-China, and Australasia. The spatial mapping concerning the Colorado River toad search terms provided additional signals from nations of three continents (the two Americas and Australia), India, the Philippines, and Europe. Several Middle Eastern and Arab countries contributed to almost one-eighth of the geographic map concerning one search term only (“*N*,*N*-dimethyltryptamine”), while Poland appeared in almost every geographic map of the online information search behavior.

Concerning the temporal mapping, web users searched the most for “*N*,*N*-dimethyltryptamine” and “5-MeO-DMT” and the least for the non-abbreviated search term “5-methoxy-*N*,*N*-dimethyltryptamine”. The strongest correlations existed between three related terms: 5-MeO-DMT and the toad species’ search terms. The same three terms exhibited a significant positive linear relationship with time.

We would also like to highlight an important issue concerning the current study as it covers countries with laws regulating access to psychoactive substances, including psychedelics, in different ways. For example, the Czech Republic law is liberal in contrast to the situation in Poland, where the law is very restrictive. Therefore, if someone in Poland browsed the web for information concerning 5-MeO-DMT, it is more likely to be for recreational rather than medical use.

### 3.2. Clinical Manifestations and Subjective Experiences

Oral ingestion of DMT after drinking ayahuasca decoction causes decreased motor activity and other neurochemical effects leading to cognitive impairment, increased prolactin and cortisol levels, and decreased lymphocyte counts [48]. Exceeding 15 or 30 times the commonly used ritual decoction doses increases serotonergic transmission [49]. Subjective experiences—depending on the route of 5-MeO-DMT application—typically include distortion of auditory and temporal perception, heightened emotional states, and a brief sense of ego dissolution, also known as “ego death” (experiencing oneness with the world or dissolution of boundaries between self and world), and experiences of mystical nature [24].

Between 2014 and 2016, a study was conducted on the subjective experiences of encountering God, both naturally and experiencing such encounters after taking classic psychedelics: psilocybin, LSD, ayahuasca, or DMT. Researchers collected data from 4285 individuals. The study found that people who consumed ayahuasca indicated the most favorable results while indicating increased life satisfaction and significant improvements in social relationships [50]. However, many of the encounters indicated as positive are at odds with the position of Ralph Metzner, an expert and researcher of South American native traditions, according to whom the experience after drinking the decoction—by contacting ancestral spirits—is more of a frightening than pleasant event for the natives [51].

### 3.3. Adverse Events, Intoxications, and Deaths

Observed side effects following the use of 5-MeO-DMT include physical effects, such as changes in blood pressure and heart rate, palpitations, chest tightness, and physical shaking (tremors), as well as psychological effects, including sadness, guilt, severe fear, anxiety, grief, and near-death experiences (NDE) [52]. In addition, anxiety, confusion, paranoia, vomiting, nausea, fatigue, loss of body perception, and sensory disturbances may occur [24]. In addition, reactivation of sensations within a week after DMT exposure has been reported to occur more frequently after vaporization than intramuscular injection [53], as well as the rare occurrence of psychotic states [24].

Results of the online Global Ayahuasca Survey collected from 2017 to 2019 with nearly 11,000 participants from more than 50 countries showed that acute physical side effects were reported by 69.9% of the sample, while adverse mental health effects in the weeks or months after ingestion were reported by 55.9% of the study participants. In the sample, nausea and vomiting were reported most frequently as adverse physical effects, while epileptic seizures, headaches, and fainting were less frequent. A small percentage of participants (2.3%) reported adverse physical health effects requiring medical attention. Of the adverse mental health effects, emotional–cognitive disorders (42.0%) and perceptual disturbances (38.3%) were reported most frequently. For most study participants, the adverse mental health effects lasted about one week. Nearly 12% reported needing professional support for the adverse effects they experienced. Interestingly, however, for most trial participants (88%), these sensations were still considered part of a positive process conducive to improving their psychological functioning [54].

The increase in demand for shamans (curanderos)—celebrants of hallucinogenic rituals—has influenced the appearance of many such offers (frequently via the Internet), usually offered by individuals without adequate training from outside the South American region. Circumstances during which a “mystical” decoction allowing contact with ancestral spirits is offered are not infrequent séances that take advantage of the gullibility of those interested. In such situations, there can be sexual abuse of those intoxicated by the brew [55], sometimes acts of unwarranted violence [56], and even fatal finales of ayahuasca rituals [57,58]. Professionals who organize rituals warn that people with high blood pressure should not use the decoction, in addition to those recovering from myocardial infarctions, diabetes, hyperthyroidism, cardiovascular disease, neurological problems, mentally ill patients, and pregnant women. In addition, it can be fatal to combine ayahuasca with alcohol, antibiotics, painkillers, cocoa, energy drinks or preservatives, and food coloring substances [27].

In the scientific literature, one can find case reports depicting both behavioral disorders caused by the use of psychedelics containing DMT in their composition: aggressive and suicidal behavior [59], as well as descriptions depicting psychiatric disorders [60]. It should be emphasized that using any psychedelic substance is not without risk, and one should always be prepared for the possibility of psychiatric complications. Such a complication, which can occur even after a single use of a psychedelic substance, is the hallucinogen persisting perception disorder (HPPD), which can impede functioning and reduce the quality of life. Usually, such a condition lasts for up to several months, but in extreme cases, it can persist for life; this complication is classified as F16 in ICD-10 and 292.89 in DSM-V [61].

### 3.4. Uses in Modern Medicine and Potentials in Precision (Personalized) Medicine

Research data suggest therapeutic-relaxing effects after taking DMT. The observed psychotherapeutic effects are indirectly related to the induction of mystical-type sensations. Therapeutic doses of 5-MeO-DMT possess neuroprotective, regenerative, and anti-inflammatory properties that may promote the treatment of cognitive impairment and PTSD [52]. A survey of 5-MeO-DMT users found that it can reduce anxiety, depression, and post-traumatic stress disorder and even effectively manage addiction [62]. These properties have been used to treat mental disorders in veterans involved in modern warfare and armed conflicts [52]. Studies conducted among ritual ayahuasca users have observed positive mental health effects, including reduced anxiety and depression, reduced use of prescription drugs, improved quality of life, better adjustment to social circumstances, easier coping with stress, and increased perceived social support [63,64]. The therapeutic use of ayahuasca decoction is addressed in numerous publications [65,66,67,68].

Particular attention is paid to the analgesic properties of the concoction and its antibacterial and antimalarial properties. Researchers reported positive results of using the brew in addiction therapy, managing cancer patients, and reducing the symptoms of Parkinson’s disease [27]. Some researchers point to reducing suicidal thoughts and behavior, which warrants further research into the potential use of psychedelics for suicide prevention [69]. Uthaug et al. (2019) stated, “*A single inhalation of vapor from dried toad secretion containing 5-MeO-DMT produces sub-acute and long-term changes in affect and cognition in volunteers*”; they concluded that based on data from 42 volunteers using dried toad secretion at several European locations [70].

Precision (personalized) medicine is an emerging disease treatment and prevention approach that considers individual variability in genes, environment, and lifestyle for each person [71]. According to the United States Centers for Disease Control and Prevention, precision medicine tailors disease prevention and treatment by considering individual differences based on a triad of genes, environment, and lifestyle [72,73]. The concept of precision medicine applies to the use of DMT in medicine, especially in psychiatry. The earlier discussion of the literature on DMT uses in modern medicine and reports on intoxications and death incidents indicate a wide variance in individuals’ responses to DMT and 5-MeO-DMT. Furthermore, micro-dosing of psychedelics might constitute an additional domain for potential personalized medicine.

In a larger cohort (n = 362), Davis et al. (2019) concluded that the 5-MeO-DMT use in a naturalistic group setting is associated with unintended improvements in anxiety and depression [74]. Davis et al. (2020) reported significant improvement concerning trauma-related suicidal ideation, PTSD, anxiety, depression, and cognitive impairment among dozens of United States special operations forces veterans [52]. Dakic et al. (2017) confirmed the occurrence of “*short-term changes in the proteome of human cerebral organoids induced by 5-MeO-DMT*” [75]. The former research data might indicate a spectrum of responses to DMT that relate to personalized medicine concepts. The researchers of the present study could foresee the successful integration of DMT into personalized (precision) psychiatry in the upcoming decades, pending a change in its current legal status.

### 3.5. Study Limitations

The current study has limitations inherent to observational study design. Hence, the level of evidence is inferior to that of experimental studies. Additionally, the present research inferred some results from digital-based rather than real-life (classical) epidemiological data because we are concerned with online information-seeking behavior. Besides, not all web users who browsed the Internet for information on DMT are actual users of the substances. Nonetheless, there could be some correlation between online search behavior and substance-use behavior. Internet users who surf the web could have used virtual private networks applications, Internet protocol (IP) masking, or implemented anonymous web browsing via the Tor browsers or DuckDuckGo search engine, among others, to conceal the actual geographic location and identity while browsing the web for critical information concerning psychedelics [76]. More advanced web users might use the deep web and darknet to seek information concerning DMT and other controlled substances, or even purchase them [77,78,79,80,81,82].

On the other hand, Google Trends does not convey raw data for each user. Instead, it conveys secondary data on the collective search behavior concerning a specific web query [46]. Data from Google Trends only reflect those with Internet access, excluding those with digital illiteracy such as some of the elderly population, disabled individuals, and the economically unprivileged. Besides, a few countries, including the People’s Republic of China (PRC), use their private Internet network, distinct web browsers, or search engines, for example, the Baidu search engine [78]. The former procedures render the search interest data from China unmappable via Google Trends. Another limitation is that we used only five search terms, including three chemical names of DMT and two generic English names of the toad. Therefore, there could be an underrepresentation of online interest in non-English-speaking countries.

## 4. Materials and Methods

The authors conducted a systematic review of the literature in international and national databases: (1) PubMed search engine and MEDLINE database of the National Library of Medicine (NLM); (2) Embase medical literature database from Elsevier; (3) the Iraqi Academic Scientific Journals (IASJ) database; (4) the University of Baghdad Digital Repository. We browsed the published literature and retrieved articles relevant to the current research by combining keywords, medical subject headings (MeSH), and Embase subject headings (Emtree) specific to DMT, 5-MeO-DMT, and the *Incilius alvarius* toad. Using Boolean operators, we also included other terms related to the history of medicine (DMT) and its uses in psychiatry and behavioral sciences.

The keywords included “5-MeO-DMT”; “5-methoxy-*N*,*N*-dimethyltryptamine”; “*Bufo alvarius*”; “Colorado River toad”; “consciousness”; “entheogens”; “epiphysis cerebri”; “ethnobotany”; “hallucinogens”; “*Incilius alvarius*”; “*N*,*N*-dimethyltryptamine”; “*N*,*N*-DMT”; “O-methyl-bufotenin”; “pineal gland”; “psychedelics”; “psychiatric disorders”; “psychiatry”; “psychoactive drugs”; “psychotomimetic”; “Sonoran Desert Toad”; “substituted tryptamines”; “the third eye”; “transcendent experiences”. Later, we scanned through the title, abstract, and full text of the retrieved articles to filter through the search results and include the final list of articles for further secondary analysis.

Concerning the infodemiology part of our study, we explored the spatial (geographic) and temporal (time-series) mapping of the online search behavior (search interest) of Internet users of the surface web via the Google Trends website. We mapped the search interest for the past 10 years (2012–2022) using 5 search terms, and these included 3 terms related to DMT and its most potent variant, the 5-MeO-DMT. The other two terms relate to the *Incilius alvarius*, the renowned amphibian toad species, which can synthesize and excrete the 5-MeO-DMT from its cutaneous and parotid glands. The five search terms are: “*N*,*N*-dimethyltryptamine”, “5-methoxy-*N*,*N*-dimethyltryptamine”, “5-MeO-DMT”, “Colorado River toad”, and “Sonoran Desert toad”. Google Trends offered time series data at monthly intervals for each search term.

We extracted the raw data from Google Trends and tabulated them using Microsoft Excel 2016, which we used to create visual illustrations of the geographic mapping of the search interest and plot the temporal mapping too. The statistician imported the raw data to IBM-SPSS version 26 to conduct descriptive statistics, bivariate correlations (Spearman rank–order correlation), and univariable (simple) linear regression to infer the trend, i.e., the temporal relationship for each search term with time. An alpha (α) value of 0.05 was considered as the cut-off margin for statistical significance.

## 5. Conclusions

Trends analyses conveyed informative spatio–temporal mapping. Internet users displayed interest in DMT, the God Molecule (5-MeO-DMT), and the Colorado River toad. Our quantitative results from Google Trends aligned with the qualitative data from the published literature. Our study highlights two main messages: (1) infodemiology and infoveillance studies are indispensable when classical epidemiological data are scarce, for example, concerning DMT use prevalence; (2) several sources of information concerning DMT exist on the Internet, including the published literature and drug forums. The information included data concerning the experimental uses of DMT for psychiatric disorders. These experimental trials indicated the potential use of DMT in precision medicine while considering an additional domain unique to psychedelics known as micro-dosing. Future research could extrapolate robust inferential evidence by conducting experimental studies of the highest level of evidence, including randomized controlled trials and reliable meta-analyses, to weigh the therapeutic benefits versus adverse events in managing psychiatric disorders on the neurotic spectrum.

## Figures and Tables

**Figure 1 pharmaceuticals-16-00831-f001:**
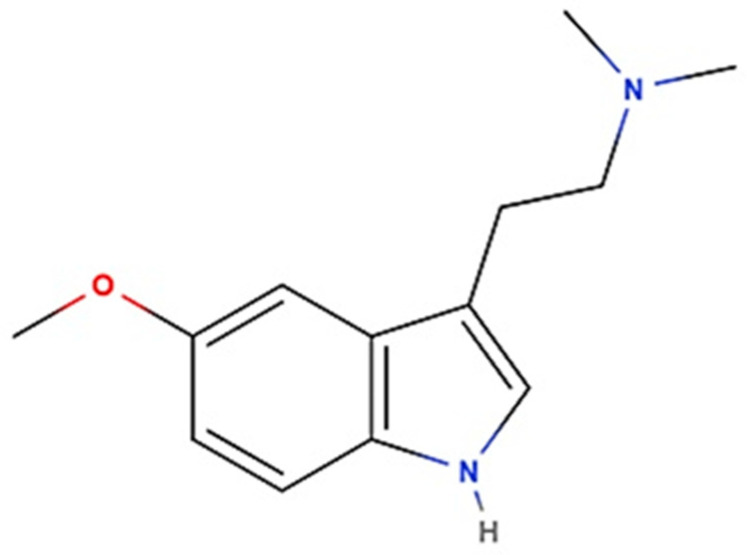
The molecular structure of 5-MeO-DMT (The molecular structure was generated using MolView free online software [20]).

**Figure 2 pharmaceuticals-16-00831-f002:**
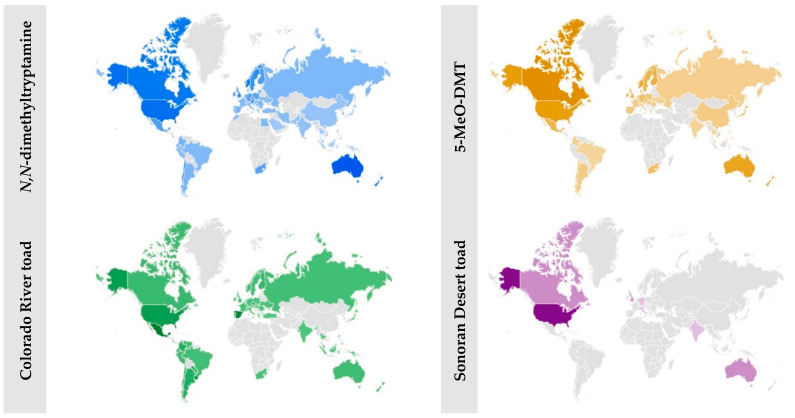
Geographic mapping of the online information search behavior worldwide. (The geographic mapping for the search term “5-methoxy-*N*,*N*-dimethyltryptamine” was absent).

**Figure 3 pharmaceuticals-16-00831-f003:**
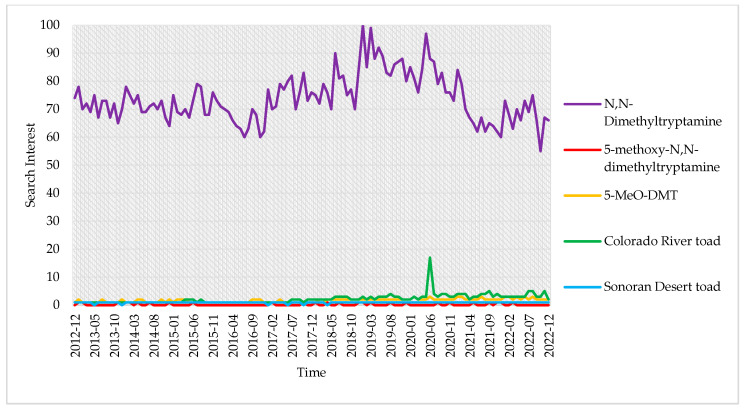
Temporal mapping of the search interest for 2012–2022. (Mapping with (upper graph) and without the search term “*N*,*N*-dimethyltryptamine” (lower graph)).

**Table 1 pharmaceuticals-16-00831-t001:** Linear trends for the search terms.

Search Term	*p*-Value	Intercept	β	95% Confidence Interval	
*N*,*N*-dimethyltryptamine	0.283	72.38	0.02	−0.02	0.07
5-MeO-DMT	<0.001	30.75	**0.37**	0.32	0.41
Colorado River toad	<0.001	2.92	**0.17**	0.13	0.21
Sonoran Desert toad	<0.001	5.15	**0.23**	0.17	0.30
5-methoxy-*N*,*N*-dimethyltryptamine	0.747	6.43	0.02	−0.09	0.12

Estimates with significant *p*-values are in bold fonts.

## Data Availability

All data are available upon reasonable request from the corresponding author within three years of the publication date.

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
