# Peer review of "Infoveillance and Critical Analysis of the Systematically Reviewed Literature on Dimethyltryptamine and the “God Molecule”"

_pharmaceuticals, 2023, doi:10.3390/ph16060831_

Round 1

Reviewer 1 Report

Review on the manuscript of Al-Imam A et al.: “Infoveillance and critical analysis of the systematically-re-2 viewed literature on Dimethyltryptamine and the "God Molecule"”.

In this manuscript, the authors reviewed the literature on N,N-Dimethyltryptamine (DMT) and related substances in international and national databases, and established a spatio-temporal mapping of internet search for DMT, 5-MeO-DMT, and the Colorado River toad. Authors concluded that Eastern Europe, the Middle East, and Far East Asia where the places with more active DMT web search. However Western Europe, Indo-China, and Australasia were more interested in the 5-MeO-DMT, and Americas, Australia, India, Philipines, and Europe registered the most abundant signals concerning the toad.

Overall, this work provides information on the legal status, risks, benefits and potential of abuse of DMT and related substances. The manuscript is very well-written and the information provided is clear. However, after a careful observation of the work, some questions and issues arise, which are presented below for consideration by the authors.

1 – The work does not cover the potential toxicity of DMT and related substances. I believe this topic should be also explored, providing a more complete understanding on the risks of DMT.  

2 – I suggest the authors to make the graphs shown in figure 3 in another software different from excel and to replace the bar colors, as there are 2 blue colors.

Author Response

Editor-in-Chief

Professor Dr. Amélia Pilar Raute

Lisboa, Portugal

Poznan – Poland, 11th April 2023

SUBJECT: COVER LETTER & COMPLIANCE WITH PEER REVIEW

Dear Prof. Dr. Amélia Pilar Raute, Editor-in-Chief of Pharmaceuticals Journal,

Dear Dr. Angel Josabad Alonso-Castro, Guest Editor at Pharmaceuticals Journal,

Dear Ms. Evelyn Du, Managing Editor of Pharmaceuticals Journal,

Dear Peer Reviewers of Pharmaceuticals Journal,

            On behalf of the coauthors, we thank the respected peer reviewers and editors for investing time and effort in critically analyzing our article. We are confident these peer reviews can enhance the manuscript's scholarly quality. We have complied with each element of the four peer review reports and revised the full-text article using the track changes option in Microsoft Word. Further, we conducted thorough proofreading following these revisions. Kindly find our response below. 

Peer-Reviewer #1

  • Review of the manuscript of Al-Imam A et al.: "Infoveillance and critical analysis of the systematically-reviewed literature on Dimethyltryptamine and the "God Molecule". In this manuscript, the authors reviewed the literature on N, N-Dimethyltryptamine (DMT) and related substances in international and national databases and established a spatio-temporal mapping of internet search for DMT, 5-MeO-DMT, and the Colorado River toad. The authors concluded that Eastern Europe, the Middle East, and Far East Asia had more active DMT web searches. However, Western Europe, Indo-China, and Australasia were more interested in the 5-MeO-DMT, and the Americas, Australia, India, Philipines, and Europe registered the most abundant signals concerning the toad. This work provides information on the legal status, risks, benefits, and potential abuse of DMT and related substances. The manuscript is very well-written, and the information provided is clear. However, after carefully observing the work, some questions and issues arise, presented below for consideration by the authors.

Response: Dear Sir/Madam, thank you for dedicating your time to analyzing our article. We addressed each element that you commented on in your report.

  1. The work does not cover the potential toxicity of DMT and related substances. I believe this topic should also be explored, providing a complete understanding of the risks of DMT.

Response: We appreciate pinpointing the lack of information concerning DMT's toxicity and adverse events. However, we already dedicated section 3.3, titled "Adverse events, intoxications, and deaths", to that. Nonetheless, we revised the section and included data concerning toxicity and side effects due to DMT use while citing new reliable bibliographic resources, including:

  • Bilhimer, M.H.; Schult, R.F.; Higgs, K.V.; Wiegand, T.J.; Gorodetsky, R.M.; Acquisto, N.M. Acute Intoxication following Dimethyltryptamine Ingestion. Case Rep. Emerg. Med. 2018, 3452691. https://doi.org/10.1155/2018/3452691.
  • Dos Santos, R.G.; Bouso, J.C.; Hallak, J.E.C. Ayahuasca, dimethyltryptamine, and psychosis: a systematic review of human studies. Adv. Psychopharmacol. 2017, 7(4), 141–157. https://doi.org/10.1177/2045125316689030.
  • WiÄ™ckiewicz, G.; StokÅ‚osa, I.; Piegza, M.; Gorczyca, P.; Pudlo, R. Lysergic Acid Diethylamide, Psilocybin and Dimethyltryptamine in Depression Treatment: A Systematic Review. Pharmaceuticals (Basel) 2021 14(8), 793. https://doi.org/10.3390/ph14080793.
  1. I suggest the authors make the graphs shown in Figure 3 in another software different from Excel and replace the bar colors, as there are 2 blue colors.

Response: Dear Sir/Madam, thank you for another keen observation. We attempted to create the line graph with IBM-SPSS, but the graph was even less appealing. We revised the line graph to the best of our "artistic" abilities. We formatted the upper limit for the y-axis, included the title for each axis, and edited the location of the legends. Further, we corrected the color coding for each line while making each visually distinct. 

Sir/Madam, we would be pleased to address any other issues you may find necessary for further revisions or corrections.

Best regards,

The corresponding author

Reviewer 2 Report

General remarks:

I have carefully reviewed the presented paper and I have multiple serious concerns about the work:

-       the Authors do not differ between reliable and unreliable sources of information. They treat as equal information gathered from the scientific journals and common newspapers – for example they cite together references 7 and 8 (line 66) although first one is Current Drug Metabolism, and the second one is Forbes Magazine. Moreover, the sentence “one of the most potent psychodelics, if not the most potent psychodelic ever” (which, by the way does not sound like taken from a scientific source) can be found in neither of the references. Other examples of using unreliable sources are the Colorado River toad picture taken from PsychonautWiki or lines 108-118, where the described manifestations are jointly referred to the Journal of Neurochemistry and PsychonautWiki. Yet, a few lines above (104-108) PsychonautWiki is cited correctly, as the easily recognizable source of plain observations. The described above mixing of reliable and unreliable sources is unacceptable in a scientific paper.

-       in lines 52-63 the Authors show the percentage results taken from the Global Drug Survey, in the way that suggest that the shown pattern is a model of a global population. The trouble is, that the Global Drug Survey authors clam on their website that their work should only be treated as a comparative source in terms of trends. Moreover, they present the demographic characteristic of the surveyed population, which is: predominantly while male, about 25 y.o. with a university degree and history of an illicit drugs use. Results taken from such a survey should be presented with the sufficient description, otherwise they are misleading.

-       the article is chaotically written. It seems that there are actually two articles in one: the review on properties of DMT and derivatives and the investigation of people’s internet behaviour. The articles should be divided.

-       the Authors claim that they have investigated “the worldwide search interest’s ... mapping”, but they have used English terms as keywords, even though they analyse both English- and non-English-speaking countries. Moreover, the Authors also draw conclusions regarding countries with non-Latin alphabet, like Russia. From my personal experience, as a person who speaks Russian, searching in the “Russian Internet” anything using the Latin alphabet produces only scarce and very adulterated results. In my opinion the Authors have actually checked the worldwide popularity of certain English terms, not the popularity of the compounds themselves. To some extent it can be probably confirmed by the overrepresentation of the English-speaking countries  in the presented maps. If the Authors claim that the investigated compounds are actually more popular in those countries, they should prove that their analytical method is immune to the language interferences.

-       the Authors do not use correctly scientific terms specific to the research field. In line 148 they claim that the potential mutagenic properties of DMT contributed to “the emergence of self-reflection, verbal communication, and problem-solving skills”. Do the Authors claim that inducing genetic mutations, which is closely related to the carcinogenic activity, could cause the above-mentioned phenomena?

-       all the binominal systematic names are to be written in italics

Therefore I consider the whole article to have basic methodological errors, which make all the resulting information not scientific.

Detailed remarks:

2.2

lines 223-227: too far-reaching conclusions, not supported by the literature data

Fig. 3: why do both graphs have the same numerical values on the y-axis? What units are presented on the y-axis? Entries?

3.4

lines 407-408: if it’s the pure compound, the source does not matter, it could be even synthetic and the results would be the same

4

line 476: the keywords should be listed

Author Response

Editor-in-Chief

Professor Dr. Amélia Pilar Raute

Lisboa, Portugal

Poznan – Poland, 11th April 2023

SUBJECT: COVER LETTER & COMPLIANCE WITH PEER REVIEW

Dear Prof. Dr. Amélia Pilar Raute, Editor-in-Chief of Pharmaceuticals Journal,

Dear Dr. Angel Josabad Alonso-Castro, Guest Editor at Pharmaceuticals Journal,

Dear Ms. Evelyn Du, Managing Editor of Pharmaceuticals Journal,

Dear Peer Reviewers of Pharmaceuticals Journal,

            On behalf of the coauthors, we thank the respected peer reviewers and editors for investing time and effort in critically analyzing our article. We are confident these peer reviews can enhance the manuscript's scholarly quality. We have complied with each element of the four peer review reports and revised the full-text article using the track changes option in Microsoft Word. Further, we conducted thorough proofreading following these revisions. Kindly find our response below. 

Peer-Reviewer #2

  • I have carefully reviewed the presented paper and have multiple serious concerns about the work.

Response: Dear Sir/Madam, we sincerely appreciate your efforts and dedication to providing critical analysis concerning our article. We aim to revise and address the issues you commented on to make the article suitable for publication.

  • The Authors do not differ between reliable and unreliable sources of information. They treat as equal information gathered from the scientific journals and common newspapers – for example, they cite references 7 and 8 (line 66), although the first is Current Drug Metabolism, and the second is Forbes Magazine. Moreover, the sentence "one of the most potent psychedelics, if not the most potent psychedelic ever" (which, by the way, does not sound like taken from a scientific source) can be found in either of the references. Other examples of using unreliable sources are the Colorado River toad picture taken from PsychonautWiki or lines 108-118, where the described manifestations are jointly referred to the Journal of Neurochemistry and PsychonautWiki. Yet, a few lines above (104-108) PsychonautWiki is cited correctly as the easily recognizable source of plain observations. The described above mixing of reliable and unreliable sources is unacceptable in a scientific paper.

Response: We appreciate your concern about using scientific (published) literature and non-scientific resources, for instance, those from drug forms and users' subjective opinions. We removed the reference material from PsychonautWiki. However, the current study is not merely a review article; it represents an infodemiology and Infoveillance study. We want to quote Eysenbach (2009): "Infodemiology can be defined as the science of distribution and determinants of information in an electronic medium, specifically the Internet, or in a population, with the ultimate aim to inform public health and public policy.".

Source: "Eysenbach G. Infodemiology and infoveillance: framework for an emerging set of public health informatics methods to analyze search, communication and publication behavior on the Internet. J Med Internet Res. 2009; 11(1): e11. doi: 10.2196/jmir.1157. PMID: 19329408; PMCID: PMC2762766.".

Therefore, we should rely on multiple information domains, including the published literature, scientific resources, and non-published data, especially those from Drug Forms, including PsychonautWiki. Besides, those drug forms include data based on the subjective experiences of psychedelics substances users, representing a cornerstone element that could not be replicated given the rarity and ethical-legal restrains of experimental studies on psychedelics. Please check these PubMed-indexed articles that specifically study or rely on exploring drug forums, including PsychonautWiki, Erowid.org, and others existing on the deep web and the darknet:

  • Pinterova N, Horsley RR, Palenicek T. Synthetic Aminoindanes: A Summary of Existing Knowledge. Front Psychiatry. 2017; 8: 236. doi: 10.3389/fpsyt.2017.00236. PMID: 29204127; PMCID: PMC5698283.
  • Wightman RS, Perrone J, Erowid F, Erowid E, Meisel ZF, Nelson LS. Comparative Analysis of Opioid Queries on Erowid.org: An Opportunity to Advance Harm Reduction. Subst Use Misuse. 2017; 52(10): 1315-1319. doi: 10.1080/10826084.2016.1276600. PMID: 28394703.
  • Catalani V, Prilutskaya M, Al-Imam A, Marrinan S, Elgharably Y, Zloh M, Martinotti G, Chilcott R, Corazza O. Octodrine: New Questions and Challenges in Sport Supplements. Brain Sci. 2018; 8(2): 34. doi: 10.3390/brainsci8020034. PMID: 29461475; PMCID: PMC5836053.
  • Al-Imam A, Santacroce R, Roman-Urrestarazu A, Chilcott R, Bersani G, Martinotti G, Corazza O. Captagon: use and trade in the Middle East. Hum Psychopharmacol. 2017; 32(3). doi: 10.1002/hup.2548. PMID: 27766667.

Following the former comprehensive discussion, we admit these resources' limitations, yet we are not strictly relying on them. Therefore, we will comply with your comments, moderate our use of those resources, and refer to their shortcomings in the limitations sub-section.

  • In lines 52-63, the Authors show the percentage results taken from the Global Drug Survey, suggesting that the shown pattern is a model of a global population. The trouble is that the Global Drug Survey authors claim on their website that their work should only be treated as a comparative source regarding trends. Moreover, they present the demographic characteristic of the surveyed population: predominantly white males, about 25 years old, with a university degree and a history of illicit drug use. Results taken from such a survey should be presented with sufficient description. Otherwise, they are misleading.

Response: Thank you for highlighting this issue. We added additional details to validate the information provided by the Global Drug Survey while also providing some confirmatory collateral data.

  • The article is chaotically written. There seem to be two articles in one: the review on properties of DMT and derivatives and the investigation of people's internet behavior. The articles should be divided.

Response: Sir/Madam, we appreciate your efforts to analyze our article systematically. Our paper contains much information, data analytics, and opinions from experts, psychedelic enthusiasts, and users who might not be considered experts by the scientific community. However, we attribute the former abundance of information to the nature of our study, which is an information surveillance "Infoveillance" study.

Therefore, we reviewed the published scientific literature in addition to surveying the psychedelics users' opinions and conducting inferential analysis based on Google Trends data concerning DMT, 5-MeO-DMT, and the toad (Colorado River toad). Accordingly, we did not only cover a narrative review of the topic but also original data. Our article might seem like two combined into one paper because the current study is an infoveillence article that covers two domains; (1) data based on published literature and non-published data, for example, those from drug forms, including PsychonautWiki; (2) the online information search behavior based on Google Trends data.

  • The Authors claim that they have investigated "the worldwide search interest's... mapping", but they have used English terms as keywords, even though they analyze both English- and non-English-speaking countries. Moreover, the Authors also draw conclusions concerning countries with the non-Latin alphabet, like Russia. From my experience as a person who speaks Russian, searching on the "Russian Internet" anything using the Latin alphabet produces only scarce and very adulterated results. In my opinion, the Authors have checked the worldwide popularity of specific English terms, not the compounds' popularity. To some extent, it can probably be confirmed by the overrepresentation of English-speaking countries in the presented maps. If the Authors claim that the investigated compounds are more popular in those countries, they should prove that their analytical method is immune to language interferences.

Response: We admit that this is a limitation of the article. However, Google Trends permits users to survey only five search terms simultaneously. Therefore, we selected those from the English language while covering the principal variants of DMT and the toad species. Besides, you suggested using the Russian language as an example, and you are correct. Accordingly, we should survey the most popular language and cover those terms in possibly the top 10 or 20% of languages worldwide, which is not feasible due to two main reasons: (1) Google Trends formerly mentioned restrictions concerning using the number of search terms; (2) We relied on using English-based terms because it represents the popular language worldwide, especially when it comes to browsing the internet. Nonetheless, we admitted those deficits concerning language and the number of search terms in the limitations section.

Last year, we published an article exploring the spatiotemporal mapping of online information search behavior concerning the most popular psychedelics. We used terms and translations from three languages only, English, Polish, and Latin. Again, the objective of the previous paper (100% concerning the online interest) differed from the current study (infoveillence). Please check our previous article here, and you will find that we used a much broader "dictionary" of terms to explore Google Trends:

Al-Imam A, Motyka MA, Witulska Z, Younus M, Michalak M. Spatiotemporal Mapping of Online Interest in Cannabis and Popular Psychedelics before and during the COVID-19 Pandemic in Poland. Int J Environ Res Public Health. 2022; 19(11): 6619. doi: 10.3390/ijerph19116619. PMID: 35682204; PMCID: PMC9180639.”.

  • The Authors do not use correct scientific terms specific to the research field. In line 148, they claim that the potential mutagenic properties of DMT contributed to "the emergence of self-reflection, verbal communication, and problem-solving skills". Do the Authors claim that inducing genetic mutations closely related to carcinogenic activity could cause the above-mentioned phenomena?

Response: We appreciate your concern and critical thinking concerning this statement in section 2.1, "The uses of DMTs in past and modern times". We want to emphasize that it was not our claim or opinion, but we paraphrased a quote from another author, Terence McKenna. Although the author might not be considered a scientist, he is regarded as a significant influencer within the psychedelics users' community. Nonetheless, we deleted all cited information concerning the stoned ape theory and Terence McKenna.

Concerning your question, "Do the Authors claim that inducing genetic mutations closely related to carcinogenic activity could cause the above-mentioned phenomena?". We answer that not all genetic mutations lead to cancers, and many mutations could be beneficial. We provide two examples from two disciplines: neurobiology and hematology. The first closely relevant example is the FOXP2 gene mutation, which enhanced cognition in primates and humans, specifically the spread of our species (Homo sapiens). Please check these other references:

  • Lieberman P. FOXP2 and Human Cognition. Cell. 2009; 137(5): 800-2. doi: 10.1016/j.cell.2009.05.013. PMID: 19490887.
  • Lakatos L, Janka Z. Az emberi agy és intelligencia evolúciója [Evolution of human brain and intelligence]. Ideggyogy Sz. 2008; 61(7-8): 220-9. Hungarian. PMID: 18763477.
  • Balaban E. Cognitive developmental biology: history, process and fortune's wheel. Cognition. 2006; 101(2): 298-332. doi: 10.1016/j.cognition.2006.04.006. PMID: 16750186.

The second example is the sickle cell trait (NOT the disease), which is a mutation that developed as a protective mechanism from a more harmful condition, the Malaria epidemic in Africa. Please check these PubMed-indexed articles:

  • Laser H, Klein R. Protection against malaria by sickle-cell trait. Biochem Soc Trans. 1977; 5(1): 292-3. doi: 10.1042/bst0050292. PMID: 892187.
  • Williams TN, Mwangi TW, Roberts DJ, Alexander ND, Weatherall DJ, Wambua S, Kortok M, Snow RW, Marsh K. An immune basis for malaria protection by the sickle cell trait. PLoS Med. 2005; 2(5): e128. doi: 10.1371/journal.pmed.0020128. PMID: 15916466; PMCID: PMC1140945.
  • Désidéri-Vaillant C, Sapin-Lory J, Di Costanzo L, Cano C, Lambrechts D, Le Mener S, Le Nestour K, Nicolas X. L'hémoglobine S protège-t-elle du paludisme ? [Does the sickle cell trait (heterozygous carrier status) confer protection against malaria?]. Med Sante Trop. 2012; 22(3): 331-3. French. doi: 10.1684/mst.2012.0091. PMID: 23174384.

On the other hand, yes, some genetic mutations affecting cancer suppressor genes and proto-oncogenes can lead to neoplasms, including benign and malignant ones. However, these mutations are essentially different, involving specific genes (cancer suppressor genes and proto-oncogenes), and could affect multiple genes or loci within the same gene. Finally, we thank you for your interesting and brain-tickling question.

  • All the binominal systematic names are to be written in italics.

Response: Thank you for noticing these errors in the writing style for the species' systematic names, and we corrected them accordingly.

  • Therefore, I consider the whole article to have basic methodological errors, which make all the resulting information not scientific.

Response: Sir/Madam, please consider our former responses and compliance with your comments. We hope we have satisfied your concerns and thoughts.

  • Section 2.2, lines 223-227: too far-reaching conclusions, not supported by the literature data

Response: We deleted these lines per your recommendations "These and other studies suggest that the "stoned monkey hypothesis," in all likelihood, is not solely the subjective conjecture of McKenna, who is fascinated by psychedelic experiences, and the studies cited above by other researchers seem to correspond closely with the hypothesis on the effect of psychedelics on the enhancement and transcendence of cognitive abilities.".

  • Fig. 3: why do both graphs have the same numerical values on the y-axis? What units are presented on the y-axis? Entries?

Response: Thank you for your vigilant attention to the graphical details. We revised the line graph and formatted the upper limit for the y-axis to 100 instead of 120. The y-axis represents the internet search interest with an ordinal scale (0 to 100). We also included the title for each axis and edited the location of the legends. Further, we corrected the color coding for each line while making it distinct from the rest. 

  • Section 3.4, lines 407-408: if it's the pure compound, the source does not matter, it could even be synthetic, and the results would be the same.

Response: We agree with your comment and revised the sentence to "Research data suggest therapeutic-relaxing effects after taking DMT.". Nonetheless, it is hypothetically possible that different variants or chemical isomers of DMT could exist in nature and differ based on the source of the substance.

  • Section 4, line 476: the keywords should be listed

Response: Thank you for noticing this issue. We included those keywords in alphabetical order: "5-MeO-DMT"; "5-methoxy-N,N-dimethyltryptamine"; "Bufo alvarius"; "Colorado River toad"; "consciousness"; "entheogens"; "epiphysis cerebri"; "ethnobotany"; "hallucinogens"; "Incilius alvarius"; "N,N-Dimethyltryptamine"; "N,N-DMT"; "O-methyl-bufotenin"; "pineal gland"; "psychedelics"; "psychoactive drugs"; "psychotomimetic"; "Sonoran Desert Toad"; "substituted tryptamines"; "the third eye"; "transcendent experiences".

Finally, we confirm that we have done comprehensive revisions all over the manuscripts, removed non-scientific references, and included new up-to-date references, including:

  • Cameron LP, Benson CJ, DeFelice BC, Fiehn O, Olson DE. Chronic, Intermittent Microdoses of the Psychedelic N,N-Dimethyltryptamine (DMT) Produce Positive Effects on Mood and Anxiety in Rodents. ACS Chem Neurosci. 2019; 10(7): 3261-3270. doi: 10.1021/acschemneuro.8b00692. Epub 2019 Mar 4. PMID: 30829033; PMCID: PMC6639775.
  • Eckernäs E, Timmermann C, Carhart-Harris R, Röshammar D, Ashton M. Population pharmacokinetic/pharmacodynamic modeling of the psychedelic experience induced by N,N-dimethyltryptamine - Implications for dose considerations. Clin Transl Sci. 2022; 15(12): 2928-2937. doi: 10.1111/cts.13410. Epub 2022 Sep 27. PMID: 36088656; PMCID: PMC9747126.
  • Calder AE, Hasler G. Towards an understanding of psychedelic-induced neuroplasticity. Neuropsychopharmacology. 2023; 48(1): 104-112. doi: 10.1038/s41386-022-01389-z. PMID: 36123427; PMCID: PMC9700802.

Sir/Madam, we would be pleased to address any other issues you may find necessary for further revisions or corrections.

Best regards,

The corresponding author

Reviewer 3 Report

It is not clear what you exactly wish to say. Your description does not follow always a logical thread. Your paper is full of information, data, and opinions, but your "systematic" search ensued in a narrative style of review of everything about N,N-Dimethyltryptamine and 5-MeO-DMT but also on Colorado River toad, also called Sonoran Desert toad (but you failed to search for Sonora Desert toad, which yields many more documents on Google); however, the narration is not consequential. It appears that the authors are anxious to report everything without a logical consequence, like if disturbing information was intruding in the flow of their writing.

In the legend of Fig. 2 you state "The geo- 278 graphic mapping for the search term "5-methoxy-N, N-dimethyltryptamine" was absent". In what sense? Was it your choice not to represent it as you did for other substances or frogs?

Amidst your Results section you state "Related web queries included those related to DMTs, purchase method, legality, forms, methods of extraction, modality of intake (use), effects, drug forums of interest (Erowid and Reddit), the toad species, and other psychoactive substances and psychedelics (MDMA, Ayahuasca, and Salvia divinorum)." This is no results, it does not present any quantitative data. Why did you put side-by-side MDMA and Salvia divinorum which have no substance related to the subject matter of your search?

Right below Discussion: where did the word "summarization" come from? I believe that your English is not bad, but you use awkward expressions and a whimsical flow of thinking that spoils your paper.

I cannot accept it in its current form. I suggest that you focus it and even split it in more than one article, but you should follow rigorous search criteria, simple and on target.

Author Response

Editor-in-Chief

Professor Dr. Amélia Pilar Raute

Lisboa, Portugal

Poznan – Poland, 11th April 2023

SUBJECT: COVER LETTER & COMPLIANCE WITH PEER REVIEW

Dear Prof. Dr. Amélia Pilar Raute, Editor-in-Chief of Pharmaceuticals Journal,

Dear Dr. Angel Josabad Alonso-Castro, Guest Editor at Pharmaceuticals Journal,

Dear Ms. Evelyn Du, Managing Editor of Pharmaceuticals Journal,

Dear Peer Reviewers of Pharmaceuticals Journal,

            On behalf of the coauthors, we thank the respected peer reviewers and editors for investing time and effort in critically analyzing our article. We are confident these peer reviews can enhance the manuscript's scholarly quality. We have complied with each element of the four peer review reports and revised the full-text article using the track changes option in Microsoft Word. Further, we conducted thorough proofreading following these revisions. Kindly find our response below. 

Peer-Reviewer #3

  • It is not clear what you exactly wish to say. Your description does not always follow a logical thread. Your paper is full of information, data, and opinions, but your "systematic" search ensued in a narrative style of review of everything about N, N-Dimethyltryptamine, and 5-MeO-DMT but also on Colorado River toad, also called Sonoran Desert toad (but you failed to search for Sonora Desert toad, which yields many more documents on Google); however, the narration is not consequential. It appears that the authors are anxious to report everything without a logical consequence as if disturbing information is intruding in the flow of their writing.

Response: Dear Sir/Madam, we appreciate your efforts to analyze our article systematically. Our paper contains cited information from the published literature, original data, experts' opinions, and subjective experiences of psychedelic enthusiasts who might not be considered experts by the scientific community. The former abundance of information relates to the nature of the current study, which is an information surveillance "Infodemiology and Infoveillance" article. We quote Eysenbach (2009): "Infodemiology can be defined as the science of distribution and determinants of information in an electronic medium, specifically the Internet, or in a population, with the ultimate aim to inform public health and public policy.".

Source: Eysenbach G. Infodemiology and infoveillance: framework for an emerging set of public health informatics methods to analyze search, communication and publication behavior on the Internet. J Med Internet Res. 2009; 11(1): e11. doi: 10.2196/jmir.1157. PMID: 19329408; PMCID: PMC2762766.

Therefore, we reviewed the published scientific literature in addition to surveying the typical psychedelics users' opinions (via drug forums, including PsychonautWiki) and conducting inferential analysis based on Google Trends data concerning DMT, 5-MeO-DMT, and the toad (Colorado River toad). We are not creating only a narrative review of the topic but also generating original data. Our article might seem like two combined into one paper because it reflects an infodemiology and infoveillence study that covers two domains; the first domain is a review of data based on published literature and non-published data, for example, those from drug forms, including PsychonautWiki; the second domain deals with the online information search behavior based on Google Trends data.

Concerning the term "Sonoran Desert toad", we already searched for it and used its more recently adopted name, "Colorado River toad". The "Sonoran Desert toad" is now considered the old generic name for what is now referred to as the "Colorado River toad", i.e., the toad species known as Bufo alvarius and Incilius alvarius. Nonetheless, we confirm using both generic terms as detailed in the Google Trends analyses section and as displayed in the results in Table 1, Figure 2, and Figure 3.

Concerning the logical flow of the article, it relates to the current study aims to review the literature concerning the DMT's past (historical) uses, legal aspects, clinical manifestations, subjective experiences, incidents of intoxications and fatalities, uses in modern medicine, and potential applications in personalized medicine, among others. We confirm that we followed the intended logical consequence detailed in the study aims.     

  • In the legend of Fig. 2, you state, "The geographic mapping for the search term "5-methoxy-N, N-dimethyltryptamine" was absent". In what sense? Was it your choice not to represent it as you did for other substances or frogs?

Response: The geographic mapping concerning the search term "5-methoxy-N, N-dimethyltryptamine" yielded no results via Google Trends. It indicates that the search interest was close to zero, which aligns with the data shown in the temporal mapping from Figure 3. In other words, the geographic mapping of "5-methoxy-N, N-dimethyltryptamine" is negligible compared to the other search terms; that's why it was absent. The lack of geographic mapping indicates that most internet users rarely use the full chemical name for the 5-MeO-DMT, which could be limited to more advanced and professional web users, including chemists, laboratory technicians, scientists, and more knowledgeable individuals.

  • Amidst your Results section, you state, "Related web queries included those related to DMTs, purchase method, legality, forms, methods of extraction, modality of intake (use), effects, drug forums of interest (Erowid and Reddit), the toad species, and other psychoactive substances and psychedelics (MDMA, Ayahuasca, and Salvia divinorum)." This is no result, and it does not present any quantitative data. Why did you put side-by-side MDMA and Salvia divinorum, which have no substance related to the subject matter of your search?

Response: We appreciate your feedback concerning the results sub-section 2.4, "Google Trends: Infodemiology and Infoveillance". We thought that including those related web queries, as generated by Google Trends, could have some value because they indicate that the same cohort of web users who searched for DMTs are simultaneously interested in web-based information concerning other subjects and substances, including drug form and other substances that are not limited to MDMA, Ayahuasca, and Salvia divinorum. However, we deleted this subsection entirely per your notes. We also deleted the sentences concerning the related web queries from the conclusions section.

  • Right below Discussion: where did the word "summarization" come from? I believe that your English is not bad, but you use awkward expressions and a whimsical flow of thinking that spoils your paper.

Response: I sincerely apologize for the incorrect use of words occasionally, as I am not a native English speaker. We revised the subsection title to "Highlights of the most important findings.".

  • I cannot accept it in its current form. I suggest you focus on it and split it into more than one article, but you should follow rigorous search criteria, simple and on target.

Response: Dear Sir/Madam, we appreciate your opinion. We confirm that we revised the article following your comments. It's not feasible at this stage to split the article as it conflicts with the opinion of the other reviewers. We attempted our best to revise the article while reconciling and addressing the comment of each of the four peer reviewers, and two of them recommended minor revisions. Once again, the infoveillence type of study forces us to consider multiple domains and resources of information on the Internet, as detailed in earlier responses. Nevertheless, we sincerely appreciate your efforts to promote the quality of our article to a superior scholarly status.

Finally, we confirm that we have done comprehensive revisions all over the manuscripts, removed non-scientific references, and included new up-to-date references, including:

  • Cameron LP, Benson CJ, DeFelice BC, Fiehn O, Olson DE. Chronic, Intermittent Microdoses of the Psychedelic N,N-Dimethyltryptamine (DMT) Produce Positive Effects on Mood and Anxiety in Rodents. ACS Chem Neurosci. 2019; 10(7): 3261-3270. doi: 10.1021/acschemneuro.8b00692. Epub 2019 Mar 4. PMID: 30829033; PMCID: PMC6639775.
  • Eckernäs E, Timmermann C, Carhart-Harris R, Röshammar D, Ashton M. Population pharmacokinetic/pharmacodynamic modeling of the psychedelic experience induced by N,N-dimethyltryptamine - Implications for dose considerations. Clin Transl Sci. 2022; 15(12): 2928-2937. doi: 10.1111/cts.13410. Epub 2022 Sep 27. PMID: 36088656; PMCID: PMC9747126.
  • Calder AE, Hasler G. Towards an understanding of psychedelic-induced neuroplasticity. Neuropsychopharmacology. 2023; 48(1): 104-112. doi: 10.1038/s41386-022-01389-z. PMID: 36123427; PMCID: PMC9700802.

Sir/Madam, we would be pleased to address any other issues you may find necessary for further revisions or corrections.

Best regards,

The corresponding author

Reviewer 4 Report

Al-Imam et al. reviewed Infoveillance and critical analysis of the systematically reviewed literature on Dimethyltryptamine and the "God Molecule. This ms is interesting and written well.

I have some minor concerns which need to be addressed to improve ms quality.

1.      The introduction part should be revised with recent literature updates.

2.      The conclusion should be revised; it is too long.

3.      The side effects of DMT should be discussed in detail.

4.      Typo errors should be addressed 

Author Response

Editor-in-Chief

Professor Dr. Amélia Pilar Raute

Lisboa, Portugal

Poznan – Poland, 11th April 2023

SUBJECT: COVER LETTER & COMPLIANCE WITH PEER REVIEW

Dear Prof. Dr. Amélia Pilar Raute, Editor-in-Chief of Pharmaceuticals Journal,

Dear Dr. Angel Josabad Alonso-Castro, Guest Editor at Pharmaceuticals Journal,

Dear Ms. Evelyn Du, Managing Editor of Pharmaceuticals Journal,

Dear Peer Reviewers of Pharmaceuticals Journal,

            On behalf of the coauthors, we thank the respected peer reviewers and editors for investing time and effort in critically analyzing our article. We are confident these peer reviews can enhance the manuscript's scholarly quality. We have complied with each element of the four peer review reports and revised the full-text article using the track changes option in Microsoft Word. Further, we conducted thorough proofreading following these revisions. Kindly find our response below. 

Peer-Reviewer #4

  • Al-Imam et al. reviewed Infoveillance and critical analysis of the systematically reviewed literature on Dimethyltryptamine and the "God Molecule. This manuscript is interesting and written well. I have some minor concerns that must be addressed to improve the manuscript's quality.

Response: Dear Sir/Madam, thank you very much for showing interest and admiring the quality of our article. We appreciate your kind words and efforts. We addressed the minor revisions you highlighted per the comments below.

  1. The introduction part should be revised with recent literature updates.

Response: We revised the introduction section and included some new information, citing recent literature updates and reliable bibliographic materials.

  1. The conclusion should be revised; it is too long.

Response: We revised and shortened the conclusions section while emphasizing coherent key messages and conclusions for the readers. Kindly find the revised conclusions section below.

"Trends analyses conveyed informative spatio-temporal mapping. Internet users displayed interest in DMT, the God Molecule (5-MeO-DMT), and the Colorado River toad. Our quantitative results from Google Trends aligned with the qualitative data from the published literature. Our study highlights two main messages: 1) Infodemiology and infoveillance studies are indispensable when classical epidemiological data are scarce, for example, concerning DMT use prevalence; 2) Several sources of information concerning DMTs exist on the Internet, including the published literature and drug forums. The information included data concerning the DMT's experimental uses for psychiatric disorders. These experimental trials indicated the potential use of DMTs in precision medicine while considering an additional domain unique to psychedelics: micro-dosing. Future research could extrapolate robust inferential evidence by conducting experimental studies of the highest level of evidence, including randomized controlled trials and reliable meta-analyses, to weigh out the DMT therapeutic benefits versus its adverse events in managing psychiatric disorders on the neurotic spectrum.".

  1. The side effects of DMT should be discussed in detail.

Response: Thank you for your insightful comment concerning the DMT's adverse effect and toxicity, which also aligns with the remarks from peer reviewer #1. However, we have already dedicated section 3.3, titled "Adverse events, intoxications, and deaths", to that. Nonetheless, we further updated and expanded it while discussing the requested details using up-to-date and new reliable references, including:

  • Bilhimer, M.H.; Schult, R.F.; Higgs, K.V.; Wiegand, T.J.; Gorodetsky, R.M.; Acquisto, N.M. Acute Intoxication following Dimethyltryptamine Ingestion. Case Rep. Emerg. Med. 2018, 3452691. https://doi.org/10.1155/2018/3452691.
  • Dos Santos, R.G.; Bouso, J.C.; Hallak, J.E.C. Ayahuasca, dimethyltryptamine, and psychosis: a systematic review of human studies. Adv. Psychopharmacol. 2017, 7(4), 141–157. https://doi.org/10.1177/2045125316689030.
  • WiÄ™ckiewicz, G.; StokÅ‚osa, I.; Piegza, M.; Gorczyca, P.; Pudlo, R. Lysergic Acid Diethylamide, Psilocybin and Dimethyltryptamine in Depression Treatment: A Systematic Review. Pharmaceuticals (Basel) 2021 14(8), 793. https://doi.org/10.3390/ph14080793.
  1. Typo errors should be addressed.

Response: We thoroughly proofread the full-text article following the revisions.

Sir/Madam, we would be pleased to address any other issues you may find necessary for further revisions or corrections.

Best regards,

The corresponding author

Round 2

Reviewer 2 Report

I am sorry to say that, but in my opinion the main weaknesses of the paper are not improved.

1. The Authors still rely on an interview presented in the Forbes magazine as the sole source of the information "The 5-MeO-DMT exists in several plant species and the Colorado River toad (Incilius alvarius), while the toad is native to northern Mexico and the southwestern United States". This interview is conducted with a person who is running a commercial project on selling DMT, "a psychedelic wellness company" as the CEO describes it. It is not a source of reliable information like the ones cited in the reviewed paper. Moreover, this very information (on presence of CMT in plants and toads) is absent in the interview. It is a serious scientific misconduct. How can we be sure, if the Authors did not perform the similar misuse to other sources, including scientific ones? This issue alone - in my opinion - is sufficient to reject the paper.

2. In their response justifying the use of  non-scientific sources, the Authors show (among the others) the papers published in Front. Psychiatry (https://doi.org/10.3389/fpsyt.2017.00236 - the first on the list). However, in the  article they refer to it is clearly stated: "For typical use and effects, Erowid, PsychonautWiki, Bluelight, and Drugs-Forum were searched". This means, that the credibility framework of sources like PsychonautWiki are relatively narrow and not all the information found there are equally reliable. The Authors clearly fail to recognize it.

3. The response of the Authors on my concern regarding the "mutagenicity" term, in which they show the examples of non-harmful mutations shows clearly that they do not understand the meaning of the term within the life-sciences research field. This, in turn, promotes drawing adulterated conclusions.

Summing up, I still consider that the paper should be rejected.

Author Response

Peer-Reviewer #2

  • The Authors still rely on an interview presented in Forbes magazine as the sole source of information "The 5-MeO-DMT exists in several plant species and the Colorado River toad (Incilius alvarius), while the toad is native to northern Mexico and the southwestern United States". This interview is conducted with a person running a commercial project on selling DMT, "a psychedelic wellness company", as the CEO describes it. It is not a source of reliable information like the ones cited in the reviewed paper. Moreover, this very information (on the presence of DMT in plants and toads) is absent in the interview. It is serious scientific misconduct. How can we be sure that the authors did not perform a similar misuse of other sources, including scientific ones? In my opinion, this issue alone is sufficient to reject the paper.

Response: Dear Sir/Madam, thank you for pointing out that Forbes magazine is not considered a credible source of information. Per your comments, we deleted and replaced the former reference with reliable scientific bibliographic material, as detailed below.

Uthaug, M.V.; Lancelotta, R.; Szabo, A.; Davis, A.K.; Riba, J., & Ramaekers, J.G. Prospective examination of synthetic 5-methoxy-N,N-dimethyltryptamine inhalation: effects on salivary IL-6, cortisol levels, affect, and non-judgment. Psycho-pharmacol. (Berlin) 2020, 237(3), 773–785. https://doi.org/10.1007/s00213-019-05414-w

  • In their response justifying the use of non-scientific sources, the Authors show (among the others) the papers published in Frontiers in Psychiatry (https://doi.org/10.3389/fpsyt.2017.00236 - the first on the list). However, the article they refer to clearly states: "For typical use and effects, Erowid, PsychonautWiki, Bluelight, and Drugs-Forum were searched". This means that the credibility framework of sources like PsychonautWiki is relatively narrow, and not all the information found there is equally reliable. The Authors fail to recognize it.

Response: Dear Sir/Madam, we appreciate your opinion concerning drug forums and psychedelics websites, including PsychonautWiki. However, we deleted that reference material and excluded PsychonautWiki following the first round of revisions. We included information while citing a publication from Frontiers in Psychiatry to highlight the importance of the drug forums in psychedelics and psychoactive substances research.

  • The response of the authors to my concern regarding the "mutagenicity" term, in which they show examples of non-harmful mutations, shows clearly that they do not understand the term's meaning within the life-sciences research field. This, in turn, promotes drawing adulterated conclusions.

Response: Dear Sir/Madam, we thank you for your feedback and appreciate your concerns about our limited understanding of the term mutagenicity. Nonetheless, we assure you that the current study's authors understand it well. The authors have good knowledge of medicine, biological sciences, and psychology. We understand that the term "mutagenicity" generally refers to the potential of a specific chemical substance to induce a genetic mutation. Therefore, we debated using two examples from the biological life sciences concerning (1) FOXP2 mutation and (2) Sickle cell trait. We also wrote in the revised version, "…the interaction between the former genomic machinery and serotoninergic psychedelics is still unknown.". The former statement indicates that it's unknown if psychedelics, as a chemical substances, could possess mutagenic properties. Nonetheless, we further revised that statement for better clarity for the readers.

Finally, please let us know if you require further amendments concerning our article. We are more than happy to address all of your requirements.

Best regards,

The corresponding author

Reviewer 3 Report

Yes, I accept. 

Author Response

Peer-Reviewer #3

  • Yes, I accept.

Response: Dear Sir/Madam, on behalf of my coauthors and colleagues, we thank you for accepting the article following rigorous peer revisions. Further, we confirm that we thoroughly proofread the full-text article following this round of minor revisions.

Finally, please let us know if you require further amendments concerning our article. We are more than happy to address all of your requirements.

Best regards,

The corresponding author
